# A Novel Synthesized 1D Nanobelt-like Cobalt Phosphate Electrode Material for Excellent Supercapacitor Applications

**DOI:** 10.3390/ma15228235

**Published:** 2022-11-19

**Authors:** S. K. Shinde, Monali B. Jalak, Swapnil S. Karade, Sutripto Majumder, Mohaseen S. Tamboli, Nguyen Tam Nguyen Truong, Nagesh C. Maile, Dae-Young Kim, Ajay D. Jagadale, H. M. Yadav

**Affiliations:** 1Department of Biological and Environmental Science, College of Life Science and Biotechnology, Dongguk University, Biomedical Campus, 32 Dongguk-ro, Ilsandong-gu, Siksa-dong, Goyang-si 10326, Republic of Korea; 2Department of Physics, Shivaji University, Kolhapur 416004, India; 3Department of Green Technology, University of Southern Denma.+8/rk, Campusvej 55, DK-5230 Odense, Denmark; 4Department of Physics, Yeungnam University, Gyeongsan 38541, Republic of Korea; 5Korea Institute of Energy Technology (KENTECH), 200 Hyeokshin-ro, Naju 58330, Republic of Korea; 6School of Chemical Engineering, Yeungnam University, 280 Daehak-Ro, Gyeongsan 38541, Republic of Korea; 7Department of Environmental Engineering, Kyungpook National University, 80 Daehak-ro, Buk-gu, Daegu 41566, Republic of Korea; 8Center for Energy Storage and Conversion, School of Electrical & Electronics Engineering, SASTRA Deemed University, Thanjavur 613401, India; 9School of Nanoscience and Biotechnology, Shivaji University, Kolhapur 416004, India

**Keywords:** Co_2_P_2_O_7_, hydrothermal method, 1D nanobelt, hybrid asymmetric supercapacitor

## Abstract

In the present report, we synthesized highly porous 1D nanobelt-like cobalt phosphate (Co_2_P_2_O_7_) materials using a hydrothermal method for supercapacitor (SC) applications. The physicochemical and electrochemical properties of the synthesized 1D nanobelt-like Co_2_P_2_O_7_ were investigated using X-ray diffraction (XRD), X-ray photoelectron (XPS) spectroscopy, and scanning electron microscopy (SEM). The surface morphology results indicated that the deposition temperatures affected the growth of the 1D nanobelts. The SEM revealed a significant change in morphological results of Co_2_P_2_O_7_ material prepared at 150 °C deposition temperature. The 1D Co_2_P_2_O_7_ nanobelt-like nanostructures provided higher electrochemical properties, because the resulting empty space promotes faster ion transfer and improves cycling stability. Moreover, the electrochemical performance indicates that the 1D nanobelt-like Co_2_P_2_O_7_ electrode deposited at 150 °C deposition temperature shows the maximum specific capacitance (Cs). The Co_2_P_2_O_7_ electrode prepared at a deposition temperature 150 °C provided maximum Cs of 1766 F g^−1^ at a lower scan rate of 5 mV s^−1^ in a 1 M KOH electrolyte. In addition, an asymmetric hybrid Co_2_P_2_O_7_//AC supercapacitor device exhibited the highest Cs of 266 F g^−1^, with an excellent energy density of 83.16 Wh kg^−1^, and a power density of 9.35 kW kg^−1^. Additionally, cycling stability results indicate that the 1D nanobelt-like Co_2_P_2_O_7_ material is a better option for the electrochemical energy storage application.

## 1. Introduction

The energy demand for industrial applications has risen steadily over the past decade [1,2], leading to the increasing exploitation of several energy sources, including solar energy, biofuels, coal, wind energy, and biomass [3,4]. However, there remain several bottlenecks in the energy industry, including limitations in the conversion and storage capacity of power plants and the transport of electricity from generation sites [2,5]. Supercapacitors (SCs) have become a vital element of the portable electronics industry to overcome these issues. Though SCs are superior to traditional batteries and capacitors in terms of their higher specific capacitance and energy density, they still suffer from poor electrical conductivity and a low power density [6,7,8]. Pseudocapacitor electrodes exhibit rapid redox and reversible redox reactions during electrochemical processes, leading to improved electrochemical performance when compared to double-layer capacitors [6,9]. Various porous nanomaterials have been used for pseudocapacitors, including transition metal oxides/hydroxides (TMO/OHs) [10,11,12], polymer-based composites [13], and C- and N-doped TMO/OHs [14,15,16], offering higher theoretical specific capacitance with better cycling stability. However, a major limitation of these nanomaterials for use in SC applications is their lower electronic conductivity and instability at higher current densities [17]. Therefore, many researchers have sought to develop novel porous electrode materials with higher electrical conductivity and stability [18]. It is well-known that a porous surface morphology can improve the electrical conductivity of electrode materials, thus there has been a particular focus on designing preparation methods that can fabricate suitable nanostructures with a high porosity [6,19]. However, it remains difficult to improve the specific capacitance of transition metal oxides/hydroxide materials. Therefore, a number of researchers have attempted to develop novel materials based on transition metal oxides to produce higher electrical conductivity and electrochemical performance [20].

In the present study, we focus on transition metal-based phosphide materials, which provide higher electrical conductivity and specific surface area [21]. Previous reports have indicated that transition metal-based phosphides are potential candidates for the SC applications [18]. In particular, cobalt-based phosphides (CoPs) are an attractive candidate as an electrode material for SC applications due to their higher electrical conductivity, eco-friendly nature, higher specific capacitance, and long-term durability. The fabrication of CoP materials involves both covalent and metallic bonds [2,6,7]. The covalent bonds in CoPs promote more rapid Faradaic reactions, which can improve electrochemical performance. Similarly, the metallic bonds in CoPs can promote more rapid ion transfer, leading to higher electrical conductivity and electrochemical performance [6,19]. This suggests that CoPs are potential electrode materials for pseudocapacitive applications [2,6,7,8]. Several electrochemical applications of CoPs with various structures have been reported in previous studies, including SCs [19,20,21,22], hydrogen evolution [23,24], water splitting [25], the oxygen evolution reaction [26], catalytic hydrogenation [27], lithium-ion batteries [28], CO_2_ reduction [29], photoelectrochemical hydrogen evolution [30], and the photocatalytic production of H_2_ and H_2_O_2_ [31]. In the present study, we focus on nanobelt-like CoPs fabricated using a simple and eco-friendly hydrothermal method, which is superior to other chemical and physical methods due to its simplicity, low cost, and lack of expensive equipment. In addition, the hydrothermal method is useful for the growth of different nanostructures via the manipulation of simple process parameters, such as the deposition temperature.

In the present investigation, we successfully synthesized three Co_2_P_2_O_7_ electrodes by changing the deposition temperature (120, 150, and 180 °C; denoted as CoP-150, CoP-120, and CoP-180, respectively) in the hydrothermal method. The effect of the deposition temperature on the crystal structure, surface morphology, and electrochemical performance of Co_2_P_2_O_7_ was studied. Highly transparent, uniform, and vertically grown nanobelts facilitate ion transfer and more rapid electrochemical properties, meaning that nanobelt-like nanostructures can play an important role in improving the properties of SCs. The CoP-150-based electrode produced better electrochemical results than did the CoP-120 and CoP-180 electrodes, which may be because the uniform and transparent nanobelt-like nanostructures provided a larger specific surface area and faster Faradaic reactions. We also fabricated a solid-state hybrid SC device using as-prepared CoP-150 as the positive electrode, activated carbon (AC) as the negative electrode, and PVA/KOH gel as the electrolyte. The CoP-150//AC device exhibited a higher Cs of 266 F g^−1^ with an energy density of 83.16 Wh kg^−1^, and a power density of 9.35 kWh kg^−1^, with excellent cycling stability up to 84% for 5000 cycles. The electrochemical results for the three-electrode and two-electrode devices demonstrated that the CoP-150 nanomaterial has significant potential as electrode material in hybrid SC devices.

## 2. Experimental Details

### 2.1. Materials

Analytical-grade chemicals were used for all experiments. Cobalt (II) nitrate hexahydrate (Co(NO_3_)_2_.6H_2_O, 99.8%), sodium phosphate (NaH_2_PO_4_, 99.8%), and activated carbon (AC, 99.8%) from Sigma-Aldrich (St. Louis, MO, USA) used without further purification.

### 2.2. Synthesis of CoP_2_O_4_ Nanobelts

To produce the 1D Co_2_P_2_O_7_ nanobelts, a simple hydrothermal method was used. The Co(NO_3_)_2_.6H_2_O and NaH_2_PO_4_ precursors were dissolved in 50 mL of deionized water with continual stirring at an equimolar concentration (0.1 M). The transparent solution was then transferred to a Teflon-lined autoclave after mixing. The hydrothermal setup was placed in a pre-programmed oven set at 120 °C for 12 h. The blue-colored product (CoP-120) was then cleaned many times in deionized water after it had cooled to room temperature. Finally, the sample was dried overnight in a vacuum at 80 °C. The CoP-150 and CoP-180 samples were prepared using the same process except with reaction temperatures of 150 °C and 180 °C, respectively.

### 2.3. Characterization

The crystalline structure of the samples was analyzed using a Rigaku X-ray diffractometer (D/Max 2500V/PC system, Japan). A Theta Probe AR-XPS system (Thermo Fisher Scientific, Oxford, UK) was used to analyze the chemical composition of the samples, while their morphology was analyzed using field-emission scanning electron microscopy (FE-SEM, SUPRA 40 VP Carl Zeiss, Jena, Germany) equipped with an energy dispersive X-ray (EDS) analyzer (Carl Zeiss, Germany). Morphological and compositional analyses were also conducted using transmission electron microscopy (TEM; H-7650 Hitachi Ltd., Tokyo, Japan).

### 2.4. Electrochemical Testing

Electrochemical measurements of the electrodes were obtained using a Versa STAT 3 electrochemical workstation with a three-electrode system. The as-prepared CoP-120, CoP-150, and CoP-180 materials were used as the working electrode, Pt wire was used as the counter-electrode, and Ag/AgCl was used as the reference electrode, with 1 M KOH solution employed as the electrolyte. Cyclic voltammetry (CV), galvanostatic charge–discharge (GCD), and electrochemical impedance spectroscopy (EIS) analyses were also conducted on the Versa STAT 3 electrochemical workstation.

### 2.5. Fabrication of the Asymmetric SC Device

A two-electrode flexible asymmetric SC device was fabricated with a positive Co_2_P_2_O_7_ electrode and commercial AC as the negative electrode. PVA/KOH gel was used as a solid-state electrolyte and printing paper was used as a separator. CV, GCD, and EIS measurements were performed using the same workstation, and the electrochemical parameters were analyzed.

#### Calculation of the Supercapacitors Parameters

The specific capacitance (Cs) of all prepared Co_2_P_2_O_7_ samples were calculated using the following Equation (1) from the CV curves [32]:(1)$$Cs=\frac{1}{m\nu (V_{c}-V_{a})}\int_{V_{a}}^{V_{c}}I(V)dV$$
where *V* and (*V_c_* − *V_a_*) are scan rates and potential window of the working electrode material, respectively.

The specific capacitance (*Cs*) of all prepared Co_2_P_2_O_7_ samples was calculated using the following Equation (2) from the GCD curves,
(2)$$Cs=\frac{I_{d}\times T_{d}}{\Delta V\times m}$$
where I_d_ and *T_d_* are discharge current and the discharge time, respectively.

## 3. Results and Discussion

### 3.1. Structural Analysis

X-ray diffraction (XRD) analysis was used to determine the phase identity, crystal structure, phrase formation, and purity of the prepared Co_2_P_2_O_7_ samples. Figure 1a presents the XRD patterns for the CoP-120, CoP-150, and CoP-180 samples prepared at various deposition temperatures. The CoP-150 sample exhibited greater diffraction intensity and sharpness than did the CoP-120 and CoP-180 samples, possibly due to its lower crystallinity. This is related to the growth process of Co_2_P_2_O_7_, which may have been affected by the deposition temperature [18,19]. Diffraction peaks were observed at 24.3, 29.72, 30.24, 35.03, 42.15, 43.24, and 47.20°, which were ascribed to the (21-2), (022), (40-2), (230), (51-3), (610), and (53-1) diffraction planes, respectively. All of these diffraction planes closely matched the standard JCPD card number (JCPDS number: 01-034-1378), indicating the presence of Co_2_P_2_O_7_ with a monoclinic crystal structure [20,21]. No other diffraction peaks were observed, suggesting that the Co_2_P_2_O_7_ was pure phase [33].

### 3.2. Elemental and Compositional Analysis

To confirm the structural properties of the Co_2_P_2_O_7_ samples, we carried out X-ray photoelectron spectroscopy (XPS) to explore the elemental composition of the material surface in more detail. Figure 1b displays the survey scan spectra for the Co_2_P_2_O_7_ samples prepared at different deposition temperatures. The full scan suggests that Co, P, and O were present in the Co_2_P_2_O_7_ samples, indicating the successful formation of a pure Co_2_P_2_O_7_ phase [6,10,31]. The core-level spectrum for Co 2p exhibited four peaks at 803.2, 797.9, 786.1, and 782.2 eV (Figure 1c) [30,31,33]. The main peaks at 782.2 and 797.9 eV were related to Co 2P_3/2_ and Co 2P_1/2_, while the two remaining sub-peaks represented satellite peaks of Co 2p [15,16,33]. These observed peaks were indicative of the formation of the Co_2_P_2_O_7_ phase. The core-level P 2p spectrum is presented in Figure 1d [29,30,31], indicating the presence of phosphide. Peaks were observed at 134.2 and 133.2 eV, which were related to P 2p_1/2_ and P 2p_3/2_, respectively [19]. Furthermore, one additional peak was observed at 133.54 eV, which was ascribed to phosphide oxides [6,18]. Figure 1e presents the core-level O 1s spectrum. The characteristic peaks at 532.7 eV and 531.5 eV indicated the presence of O in Co_2_P_2_O_7_. The main peak at 532.7.4 eV was related to the C-O-P molecule. An additional peak was observed at 531.5 eV, which was attributed to the double bonds of the C=O and P=O molecules [33,34]. Taken as a whole, the XPS results indicated the successful formation of a pure Co_2_P_2_O_7_ phase. FT-IR results are shown in Figure 1f; the main absorption peaks at 561 cm^−1^ are related to the Co-P-O composition, and other absorption peaks at 3475 cm^−1^ indicate the presence of the OH molecule because of the whole preparation of the Co_2_P_2_O_7_ materials in the water. The FT-IR analysis confirmed the formation of the cobalt phosphate material.

### 3.3. Surface Morphology Analysis

Figure 2a–f presents the surface morphology of the as-prepared Co_2_P_2_O_7_ samples at various deposition temperatures using the hydrothermal route. For the initial deposition temperature of 120 °C, small nanosheet-like structures were observed on the surface of the sample in the SEM images (Figure 2a). However, the entire surface of the CoP-120 sample was not fully covered by these structures, which indicates that this reaction temperature was not sufficient for the uniform formation and full coating of CoP nanostructures. When the deposition temperature was increased to 150 °C, the SEM images clearly show the uniform development of unique 1D nanobelts on the surface of the CoP-150 sample (Figure 2d). At a higher magnification, the CoP-150 surface was fully covered by vertical nanobelts. Each nanobelt was separated by empty space, allowing more rapid pathways for ion exchange between the redox reactions [33,34,35]. This nanostructure also offers a higher surface area and higher electrical conductivity, thus potentially improving the electrical properties and long-term cycling stability of SC applications.

At the highest deposition temperature (180 °C), the SEM images revealed aggregations of vertical nanosheets of the CoP-180 surface [33,36,37], which had the potential to reduce the ion exchange rate and generate longer pathways for electrochemical reactions. Therefore, this surface morphology may not be useful for electrochemical applications because the ions cannot move rapidly from one medium to another [31,33,34,35,36,37,38]. The SEM analysis thus confirmed that the deposition temperature affected the surface morphology of the hydrothermally prepared CoP samples [39]. Furthermore, EDS analysis was used to confirm the elements present in the prepared Co_2_P_2_O_7_ samples, as shown in Figure 2g–i. The EDS analysis confirmed the presence of Co, P, and O in all Co_2_P_2_O_7_ samples, indicating the formation of pure-phase Co_2_P_2_O_7_ without any other impurities. The SEM and EDS results illustrated that the hydrothermal method is suitable for the preparation of Co_2_P_2_O_7_ [31,33,40].

TEM was used to further examine the surface morphology and porosity of the as-prepared Co_2_P_2_O_7_ nanobelts. Appendix A and Figure 3 present TEM images, SEAD patterns, and elemental mapping of the CoP-120, CoP-180, and optimized CoP-150 samples, respectively. Figure 3a–d shows TEM images of the optimized CoP-150 sample at lower and higher magnifications, respectively. The TEM images illustrate the completely perpendicular alignment along the horizontal axis of nanobelts with a width of 400–450 nm and a length of 1–1.2 µm. The individual nanobelts thus had a surface area of around 0.015 cm, which suggests that the CoP-150 sample would produce superior electrical performance and lower electrical induction [30,31,33,41,42,43]. The high-magnification TEM image revealed that the nanobelts had empty space on all sides, which indicated that ions could be transferred rapidly from the surface of one nanobelt to another. The interplanar spacing distance was calculated to be 0.34 nm, which was ascribed to the (21-2) crystal plane presented in the high-magnification TEM images of CoP-150 (Figure 3d) [5]. The SAED patterns revealed dark diffraction rings, which were indicative of the polycrystalline nature of the optimized CoP-150 (Figure 3e) [44].

To investigate the elemental composition of the sample in more detail, elemental mapping was employed on the CoP-120, CoP-150, and CoP-180 samples. The uniform and homogeneous distribution of Co, P, and O was observed on the surface of the samples, which indicated the pure and uniform formation of Co_2_P_2_O_7_ (Figure 3f–i) [43,44,45]. Appendix A present TEM images, SAED patterns, and elemental mapping for the CoP-120 and CoP-180 samples prepared using the hydrothermal method. The TEM images of CoP-120 revealed the non-uniform growth of the nanostructures, which indicates the deposition temperature of 120 °C was insufficient to complete the growth of the nanostructures (Appendix A) [6,33,34]. At a higher deposition temperature at 180 °C, the TEM images illustrated the aggregation of the nanobelts, which was indicative of the overgrowth of the nanostructures (Appendix A). The elemental mapping revealed that Co, P, and O were present on the surface of the CoP-120 and CoP-180 samples, which suggested the pure and uniform formation of Co_2_P_2_O_7_ (Appendix A) [43,44,45]. The surface morphology results indicate that the deposition temperature plays an important role in the formation and development of nanobelt-like nanostructures using the hydrothermal method.

BET analysis was performed to investigate more details about the porosity and specific surface areas of the sample. Figure 4 shows the N_2_ adsorption–desorption isotherms of the optimized Co_2_P_2_O_7_ sample deposited at a deposition temperature of 150 °C, and the insets show their pore-size distribution of the Co_2_P_2_O_7_ sample. The specific surface area of the optimized Co_2_P_2_O_7_ sample was obtained from BET to be 63.45 m^2^ g^1^, which is better than the previously reported values [35,36]. The optimized Co_2_P_2_O_7_ sample provided a higher surface area, which may be due to the vertical form of a nanobelt-like nanostructure with a pore radius of 18 nm. This type of surface morphology is more useful for electrochemical applications [35,36,37].

### 3.4. Electrochemical Analysis of the Co_2_P_2_O_7_ Electrodes

The SC properties of the as-prepared CoP-120, CoP-150, and CoP-180 electrodes were assessed in a three-electrode system with a 1 M KOH electrolyte. Figure 5a–c presents the CV curves for the CoP-120, CoP-150, and CoP-180 electrodes, respectively, at scan rates from 5 to 100 mV s^−1^. The CV curves for CoP-150 were noticeably different from those for CoP-120 and CoP-180, indicating that the surface morphology and deposition temperature affected the electrochemical properties [2,5,6]. The CV curves for the CoP-120 and CoP-180 electrodes were similar to each other, possibly due to the insufficient deposition temperature and overly high deposition temperature, respectively [17,18]. The CV curve for the CoP-150 electrode had a larger area under the curve and a stronger redox pair than did those for the other two electrodes, which indicates that the CoP-150 offered a stronger electrochemical performance [46]. Figure 5d presents the Cs values for the CoP-120, CoP-150, and CoP-180 electrodes at various scan rates within a potential window of 0–0.4 V. As expected, the CoP-150 electrode exhibited higher Cs than the other two electrodes (1057, 1766, and 856 F g^−1^ for the CoP-120, CoP-150, and CoP-180 electrodes, respectively, at a constant scan rate of 5 mV s^−1^), which was due to the uniform growth of the porous nanobelts [6,33,44,45,46], facilitating the ion transfer between the nanostructures and improving the electrochemical results for the CoP-150 electrode [33,34,35,36,37,39].

Another important parameter for SC performance is GCD. GCD analysis was thus used to calculate the Cs for the Co_2_P_2_O_7_ electrodes at various current densities. Figure 6a–c presents the GCD measurement curves for the CoP-120, CoP-150, and CoP-180 electrodes, respectively, at various current densities at a constant potential window. The CoP-150 electrode had a superior charge–discharge time compared to the CoP-120 and CoP-150 electrodes at a constant current density, indicating that the optimized CoP-150 electrode had improved electrochemical properties [33,40,41,42,43]. The Cs were calculated to be 1426, 1667, and 1186 F g^−1^ for the CoP-120, CoP-150, and CoP-180 electrodes, respectively, at a constant current density of 3 mA cm^−2^ (Figure 6d). The obtained Cs values for the CoP-150 electrode were 1667, 1469, 1256, 1140, 1050, 972, and 974 F g^−1^ at current densities from 3 to 21 mA cm^−2^, respectively. The main reason for the improved electrochemical properties of the optimized CoP-150 electrode is the surface morphology [33,34,35,36,37,38,39,40,41,42,43]. The vertical growth of the nanobelts on the surface generated open space between any two sheets in four directions, which facilitates the transfer of ions from one medium to another, improving the electrochemical performance [6,33,44,45,46]. Conversely, the lower electrochemical performance of the CoP-120 and CoP-180 electrodes was due to incomplete growth and aggregation of the nanostructures, respectively [17,18]. The GCD results were thus in line with the structural and surface morphological properties of the CoP-150 electrode. The cycling stability is another important parameter for electrode materials. The cycling stability of as-prepared CoP-120, CoP-150, and CoP-180 electrodes was examined at a constant scan rate of 100 mV s^−1^ in a 1 M KOH electrolyte within the same experimental parameters (Appendix A). The CoP-120, CoP-150, and CoP-180 electrodes had a capacity retention rate of 86, 93, and 82%, respectively, confirming that the CoP-150 electrode is more stable because of its stability and porous surface morphology [2,5,6,17].

To understand the electrochemical mechanisms associated with the prepared electrodes and 1 M KOH electrolyte, electrochemical impedance spectroscopy (EIS) was conducted at a constant frequency between 1−100 kHz. Figure 7 shows the Nyquist plots for the CoP-120, CoP-150, and CoP-180 electrodes, with the inset showing a magnified version of the marked area and the equivalent circuit, respectively. The solution resistance (R_s_) was calculated using the fitted equivalent circuit method. The R_s_ values for the CoP-120, CoP-150, and CoP-180 electrodes are 5.90, 4.69, and 6.69 Ω, respectively. The results indicate that the CoP-150 electrode had a lower resistance than the CoP-120 and CoP-150 electrodes, which suggests that the CoP-150 electrode is more suitable for electrochemical applications [46,47]. This is because this electrode offers a higher surface area and more rapid ion transfer, which is useful for improving SC performance [33].

### 3.5. Solid-State Asymmetric Hybrid SC

We constructed a two-electrode solid-state hybrid SC device using the optimized CoP-150 material as the positive electrode, and AC as the negative electrode, with KOH/PVA gel used as the electrolyte. Figure 8a presents the CV measurements for the Co_2_P_2_O_7_//AC SC within various potential windows at 50 mV s^−1^. The CV curves revealed that the active potential window was fixed between 0.0 to 1.5 V for the asymmetric device (Figure 8a). Figure 8b presents the CV curves for the Co_2_P_2_O_7_//AC SC at different scan rates from 5–100 mV s^−1^, respectively. The CV curves exhibited similar positive and negative current densities, which may be due to the presence of Faradaic reactions [19,20,21]. Figure 8c presents the Cs calculated for different scan rates using the solid-state asymmetric SC device. The Cs were 266, 212, 199, 178, 163, and 158 F g^−1^ at scan rates from 5 to 100 mV s^−1^, respectively. The CV curve shows that, when the scan rate increased from 5 to 100 mV s^−1^, the current and rectangular shape increased, which indicates that the Co_2_P_2_O_7_//AC device offers long-term stability and more beneficial electrochemical properties. The optimization of the potential window using GCD testing at a constant current density is presented in Figure 8d. Figure 8e presents the GCD curves of the Co_2_P_2_O_7_//AC device at current densities from 5–30 mA cm^−2^. Analysis of the discharge time revealed very small drops in Ir at lower current densities, which indicates that the SC had higher cycling stability and a higher retention capacity. Figure 8f presents the Cs calculated from the GCD curves at current densities from 5–30 mA cm^−2^. The Cs for the Co_2_P_2_O_7_//AC device were 267 F g^−1^ at a current density of 5 mA cm^−2^. Long-term durability and retraction capacity are also important parameters for solid-state SCs for practical electrical devices and electric motor applications [48]. Figure 8g presents the cycling stability of the Co_2_P_2_O_7_//AC device based on GCD curves at a constant current density of 5 mA cm^−2^, with the inset presenting the first and last GCD and CV cycles within the same potential window. The Co_2_P_2_O_7_//AC device exhibited a good retention capacity of up to 91% after 5000 GCD cycles; the inset shows the CV and GCD curves after cycling performance. Figure 8g shows that ternary Co_2_P_2_O_7_ nano-compounds have outstanding long-term cycling stability, which suggests that these electrodes are useful for electrochemical applications as positive electrode material.

Two important parameters for two-electrode devices are energy and power density. Figure 8h presents Ragone plots for the Co_2_P_2_O_7_//AC SC device in terms of its energy and power density. The highest energy density obtained was 83.16 Wh kg^−1^ at a power density of 9.35 kWh kg^−1^, which is higher than that reported for other SC devices based on ternary Co_2_P_2_O_7_ materials [46,47,48,49,50,51,52,53]. This indicates that Co_2_P_2_O_7_ materials are superior to other ternary compounds in terms of their electrochemical performance [47,48,49,50,51,52,53]. The outstanding electrochemical properties of the solid-state asymmetric Co_2_P_2_O_7_//AC SC device can be ascribed to its higher electrical conductivity due to its lower R_s_ and R_ct_. Figure 8i presents the Nyquist plots of the Co_2_P_2_O_7_//AC SC device before and after cycling stability testing. As expected, R_s_ and R_ct_ were similar before and after cycling stability testing (1.9 and 6 Ω and 35 and 55 Ω, respectively). The difference between the first and last cycle was thus very small, which indicates that the Co_2_P_2_O_7_//AC SC device offers a higher active area and rapid electron and ion transport [19,22,23,24].

In addition, the two-series arrangement of the developed ultradurable full solid-state device demonstrated its commercial legitimacy by allowing the assembly of 37 red light-emitting diodes (LEDs) to work for more than two minutes, and this was accomplished with only thirty seconds of charging, as shown in Figure 9. The initial glow of the LEDs was quite powerful, and it endured for a long time, as can be seen from the digital images in Figure 9. The glowing LED and motor fan indicate that the Co_2_P_2_O_7_ electrode materials are the best candidates for energy storage applications.

## 4. Conclusions

In summary, we have successfully prepared 1D Co_2_P_2_O_7_ nanobelt-like material using a simple and cost-effective hydrothermal method under various deposition temperatures and its supercapacitor properties. The structural, morphological, and electrochemical results indicated that the deposition temperatures affected the supercapacitive properties of the Co_2_P_2_O_7_. The XRD and FT-IR results confirmed the formation of a pure Co_2_P_2_O_7_ phase. The surface morphological results revealed that the CoP-150 electrode had a higher surface area than the CoP-120 and CoP-180 electrodes, which is useful for improving the SC properties. The CoP-150 electrode shows a maximum Cs of 1766 F g^−1^, which is higher than that of the CoP-120 (1057 F g^−1^) and CoP-180 (856 F g^−1^) electrodes at a constant scan rate of 5 mV s^−1^, respectively. The constructed hybrid asymmetric supercapacitor device of CoP-150//AC shows the highest Cs of 266 F g^−1^, with an excellent energy density of 83.16 Wh kg^−1^ and a power density of 9.35 kW kg^−1^, respectively. The practical application of the CoP-150//AC SCs device also indicated that phosphate-based nanomaterials are the best option for electrochemical applications.

## Figures and Tables

**Figure 1 materials-15-08235-f001:**
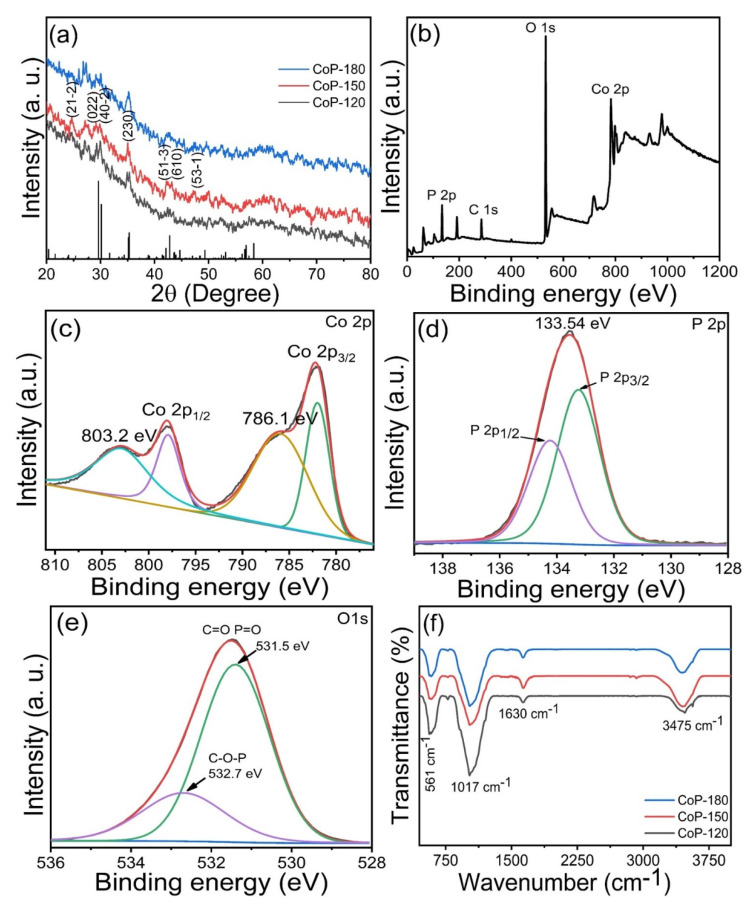
(**a**) XRD patterns of the Co_2_P_2_O_7_ materials synthesized at various deposition temperatures of 120, 150, and 180 °C, (**b**) survey spectrum of the optimized Co_2_P_2_O_7_ materials at deposition temperatures of 150 °C, (**c**–**e**) core level of the Co 2p, P 2p, and O 1s, respectively, (**f**) FT-IR spectra of the Co_2_P_2_O_7_ materials synthesized at various deposition temperatures.

**Figure 2 materials-15-08235-f002:**
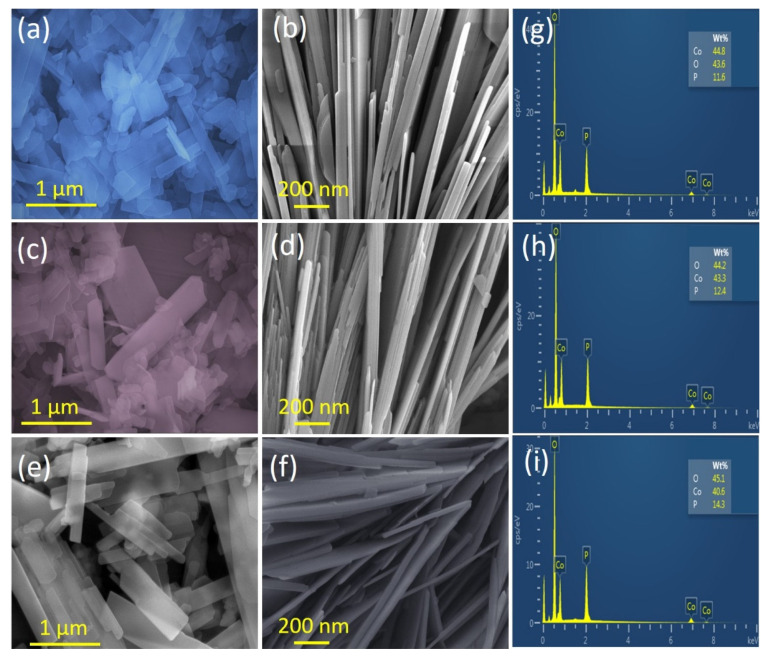
(**a**–**f**) SEM images of the Co_2_P_2_O_7_ materials synthesized at various deposition temperatures of 120, 150, and 180 °C, respectively, (**g**–**i**) EDS of the Co_2_P_2_O_7_ materials synthesized at various deposition temperatures of 120, 150, and 180 °C, respectively.

**Figure 3 materials-15-08235-f003:**
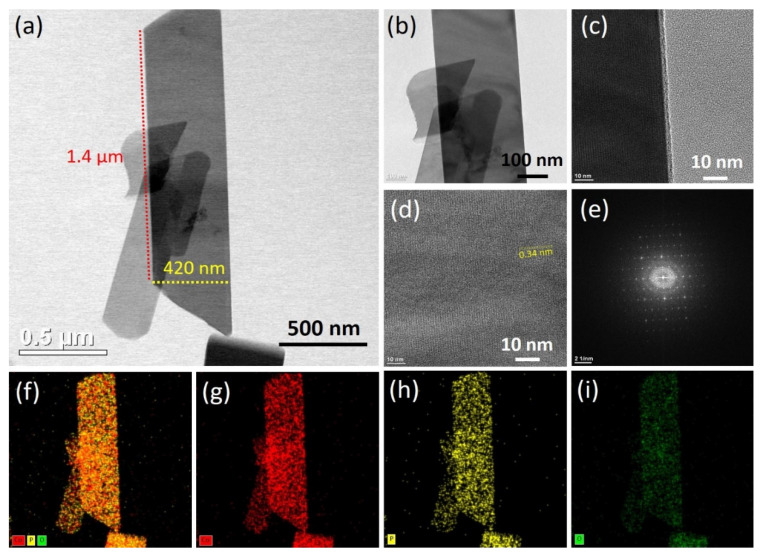
(**a**–**d**) TEM images of the Co_2_P_2_O_7_ materials synthesized at optimized deposition temperatures of 150 °C, respectively, (**e**) SAED pattern, (**f**–**i**) elemental mapping of the optimized Co_2_P_2_O_7_ materials, respectively.

**Figure 4 materials-15-08235-f004:**
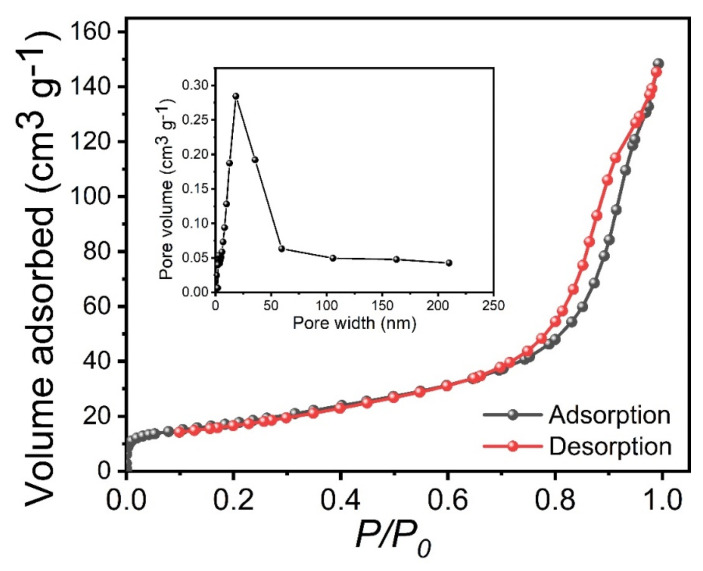
BET of the optimized Co_2_P_2_O_7_ materials at deposition temperatures of 150 °C, inset shows the pore-size distribution of the Co_2_P_2_O_7_ sample.

**Figure 5 materials-15-08235-f005:**
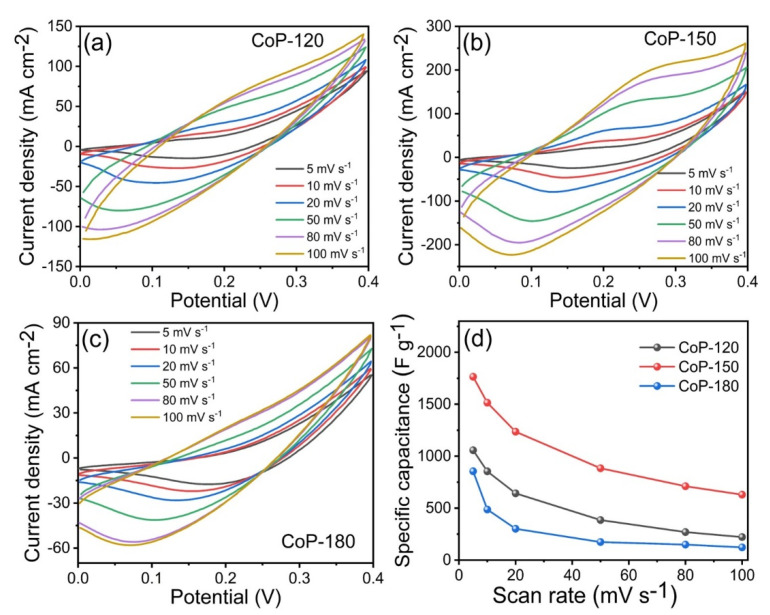
(**a**–**c**) CV measurements of the CoP-120, CoP-150, and CoP-180 electrodes at various scan rates in the range of 5–100 mV s^−1^, (**d**) specific capacitance of the CoP-120, CoP-150, and CoP-180 electrodes at various scan rates, respectively.

**Figure 6 materials-15-08235-f006:**
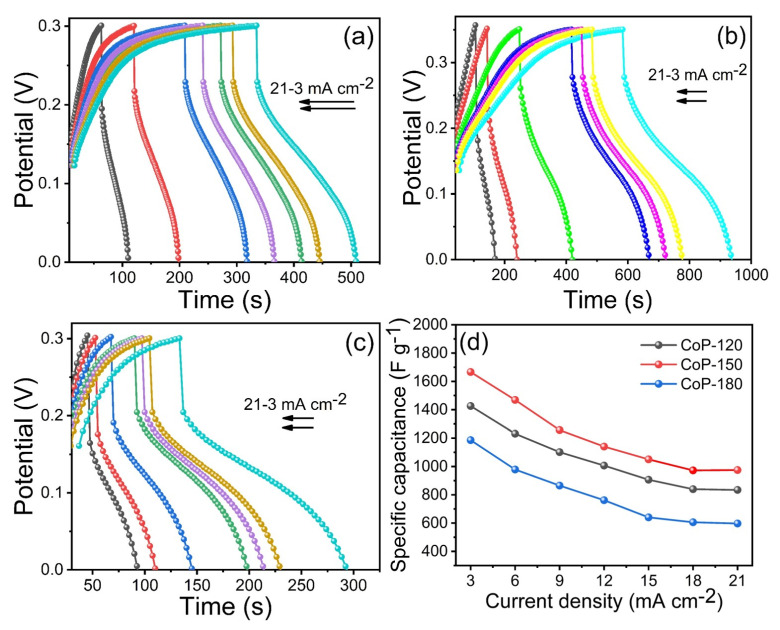
(**a**–**c**) GCD measurements of the CoP-120, CoP-150, and CoP-180 electrodes at different current densities from 3–21 mA cm^−2^, (**d**) specific capacitance of the CoP-120, CoP-150, and CoP-180 electrodes at different current densities, respectively.

**Figure 7 materials-15-08235-f007:**
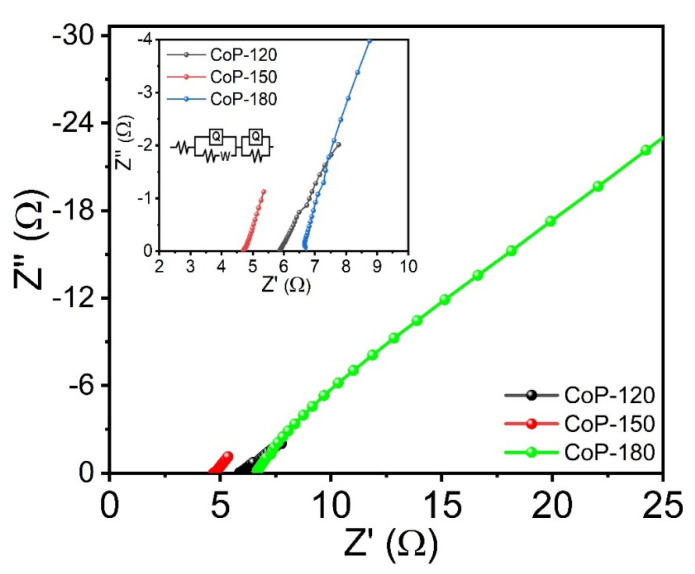
EIS plots of the Co_2_P_2_O_7_ materials synthesized at various deposition temperatures of 120, 150, and 180 °C, respectively.

**Figure 8 materials-15-08235-f008:**
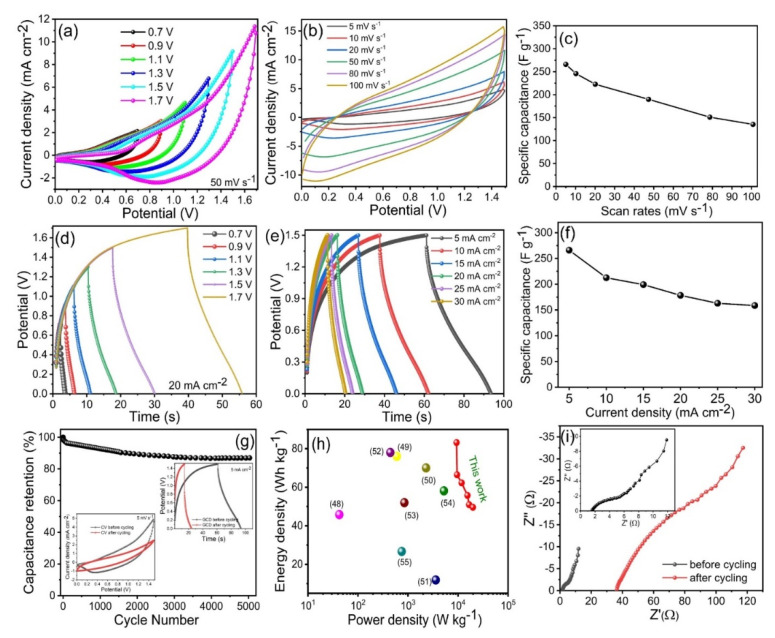
(**a**) CV curves of the asymmetric hybrid Co_2_P_2_O_7_//AC SC device at a different potential window at a constant scan rate of 50 mV s^−1^, (**b**) CV curves of hybrid Co_2_P_2_O_7_//AC SC device at a constant potential window 1.5 V, with various scan rate 5–100 mV s^−1^, (**c**) specific capacitance of the hybrid Co_2_P_2_O_7_//AC SC device at various scan rates, (**d**) GCD curves of the hybrid Co_2_P_2_O_7_//AC SC device at a different potential window at a constant current density of 20 mA cm^−2^, (**e**) GCD curves of the hybrid Co_2_P_2_O_7_//AC SC device at a constant potential window 1.5 V, with different current densities 5–30 mA cm^−2^, (**f**) specific capacitance of the hybrid Co_2_P_2_O_7_//AC SC device at different current densities 5–30 mA cm^−2^, (**g**) cycling stability of the hybrid Co_2_P_2_O_7_//AC SC device up to 5000 cycles, inset shows the CV and GCD curves of the lower scan rate (5 mV s^−1^) and current density (5 mA cm^−2^), (**h**) Ragone plot, (**i**) EIS of the hybrid Co_2_P_2_O_7_//AC SC device before and after cycling stability testing.

**Figure 9 materials-15-08235-f009:**
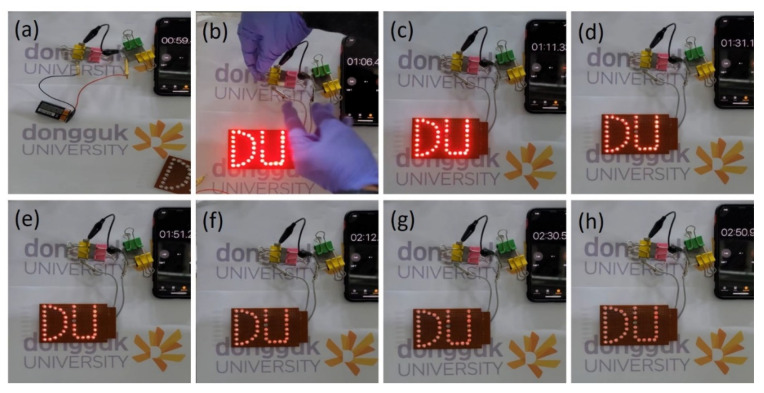
(**a**–**h**). Commercial practical demonstration of the LED with different discharging times.

## Data Availability

All the data used in this study is already included in the manuscript.

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
