# Peer review of "A Novel Synthesized 1D Nanobelt-like Cobalt Phosphate Electrode Material for Excellent Supercapacitor Applications"

_materials, 2022, doi:10.3390/ma15228235_

Round 1

Reviewer 1 Report

The genesis of Monali B. Jalak1’s paper is to demonstrate a synthetic route to prepare cobalt phosphate, and its potential functional usage as electrodes material in the supercapacitor. However, this similar strategy has been widely reported. The novelty should be provided in the manuscript.

Some contents that is doubtable are listed below.

1. There are some unit errors that needs to be rechecked. For example, it should be 5 mV s-1 in the line 10 of abstract.

2. In the introduction part, the comparison between metal phosphides and metal oxides should be explicated. Why are metal phosphides superior to metal oxides for supercapacitor application?

3. In the three-electrode system, Ag/AgCl is not recommended in the alkaline system.

4. It is recommended to put the standard Co2P2O7 on Figure 1a.

5. In the Figure 1c, all the words should be in the same direction.

6. Why is the potential window of GCD curves only 0.3V?

7. The source of activated carbon should be provided.

8. In 3.6 part, the Cs were calculated as 266, 212, 199, 178, 163, and 158 mAh−1. However, in the abstract, 266 F g−1 was denoted. The calculation method should be also provided in the experimental part.

To put my inputs together, the revised the manuscript is not acceptable. Some polish in writing could be made to make it clearer and crisper.

9. Figure S3 is missing.

10. Table 1 is missing.

Author Response

Date: 2022-10-19

Response to the Reviewers Comments

Dear Editor,

Materials,

Greetings:

Response letter for manuscript number: R1 (Materials-1958879).

Dear Editor,

This response letter accompanies the online submission of our revised manuscript (Manuscript number: materials-1958879) entitled “A novel synthesized 1D nanobelts-like cobalt phosphate electrode material for excellent supercapacitors application” by S. K. Shinde*, Monali B. Jalak, Swapnil S. Karade, Sutripto Majumder, Mohaseen S. Tamboli, Nguyen Tam Ngu-yen Truong*, Nagesh C. Maile, D. Y. Kim, and Ajay D. Jagadale, for the publication in Materials. We sincerely appreciate the editor and reviewers for reviewing and suggesting valuable comments to improve our manuscript. Following your revision letter, we have thoroughly revised our manuscript to accommodate all the comments and issues raised by the editor and reviewers. We have made substantial modifications to address the reviewer’s suggestions and comments as well as other changes that we thought appropriate to improve our manuscript. All the revisions are accordingly highlighted in this revised manuscript for the publication in  Materials.

Please feel free to contact us for any further information.

Dr. Prof. Nguyen Tam Nguyen Truong

                    M.Sc. Ph. D.               

School of Chemical Engineering, Yeungnam University, 280 Daehak-Ro, Gyeongsan 38541, Republic of Korea

Response letter to reviewers

Reviewer # 1: Comments and Suggestions for Authors Reviewer 1,

Comments and Suggestions for Authors

The genesis of Monali B. Jalak1’s paper is to demonstrate a synthetic route to prepare cobalt phosphate, and its potential functional usage as electrodes material in the supercapacitor. However, this similar strategy has been widely reported. The novelty should be provided in the manuscript.

Some contents that is doubtable are listed below.

The authors are very thankful to the reviewer for appreciating and approving our work. We have addressed all the comments and revised the manuscript thoroughly. The suggested changes have been incorporated into the revised manuscript and the relevant changes are highlighted.

  1. There are some unit errors that needs to be rechecked. For example, it should be 5 mV s-1in line 10 of the abstract.

Response to the Reviewer’s comments:

Thank you for your comment. We have corrected the values and units of specific capacitance in the revised manuscript

  1. In the introduction part, the comparison between metal phosphides and metal oxides should be explicated. Why are metal phosphides superior to metal oxides for supercapacitor application?

Response to the Reviewer’s comments:

Thank you for your comment. As per the reviewer’s suggestion, the introduction section has been thoroughly revised in order to improve the quality of the manuscript.

  1. In the three-electrode system, Ag/AgCl is not recommended in the alkaline system.

Response to the Reviewer’s comments:

Thank you for the reviewer’s comment. The authors agree with the reviewer’s comment. We have added the details of the Ag/AgCl reference electrodes in the revised manuscript, and its detailed explanation is included in section 2.4 on Page no. 3. In this paper, we use a chemically synthesized CoPO4 nanobelts-like electrode as the working electrode, the AgCl as a reference electrode, and platinum electrode as counter electrode for the supercapacitor testing. In our previous work, we have frequently used Ag/AgCl as a reference electrode for supercapacitor applications. Also, literature shows that the Ag/AgCl reference electrode has been used widely for supercapacitor testing.

  1. It is recommended to put the standard Co2P2O7on Figure 1a.

Response to the Reviewer’s comments:

Thank you for your valuable suggestion. As per the reviewer’s suggestion, we have added the standard patterns of the XRD in Figure 1a in the revised manuscript, and its detailed explanation is included in Figure 1a.

  1. In Figure 1c, all the words should be in the same direction.

Response to the Reviewer’s comments:

Thank you for your valuable suggestion. As per the reviewer’s suggestion, we have revised Figure 1c in the revised manuscript, and its detailed explanation is included in Figure 1c.

  1. Why is the potential window of GCD curves only 0.3V?

Response to the Reviewer’s comments:

Thank you for your comment. We appreciate the reviewer’s comment on the potential window of GCD curves lower than the CV curves of the Co2P2O7 electrodes. We agree with your opinion. In this article, we have attempted to achieve the wide potential window of the Co2P2O7 electrode having different nanostructures. All the Co2P2O7 electrodes during charging (Fig. 6) can reach the upper potential limit of 0.4 V when the applied current density was higher than 50 mA cm-2. This implied that the Co2P2O7 electrodes can truly act as supercapacitive electrodes having both high power and energy density. More particularly, the high power results from the high discharge current densities whereas the high energy at the corresponding current densities originates due to the wide potential window. On the other hand, for current densities less than 21 mA cm-2, the upper potential limit was not reached which may be due to the inability of the inner active sites of the electrode to maintain redox transitions. At low current densities. This can be explained as follows when the first layer of the active sites on the Co2P2O7 electrode undergoes redox transitions, they develop their potential at the interface and for the next level of transitions, the other electrodes have to surpass this potential to further oxidize the electrode surface. Nevertheless, such electrodes can be used in applications where high energy density is required. Thus, our intention is to expose our electrodes to both high-power and energy application.

The voltage drop results from the ohmic resistance of the entire electrochemical cell. It is the series combination of the three different types of resistances namely intrinsic resistance of the electrode, contact resistance between the electrode and current collector, and ionic resistance of the electrolyte. Further efforts are ongoing to minimize both the intrinsic and contact resistances.

Reference

[1] G. J. Navathe, D. S. Patil, P. R. Jadhav, D. V. Awale, A. M. Teli, S. C. Bhise, S. S. Kolekar, M. M. Karanjkar, J. H. Kim, P. S. Patil, J.  Electroanal. Chem., 738 (2015) 170–175

[2] S. D. Jagadalea, A. M. Teli, S. V. Kalake, A. D. Sawant, A. A. Yadav, P. S. Patil, J. Electroanal. Chem., 816 (2018) 99–106

  1. The source of activated carbon should be provided.

Response to the Reviewer’s comments:

Thank you for your comment. As per the reviewer’s suggestion, we have revised the material details section and added the source of activated carbon in the section in the revised manuscript.

  1. In 3.6 part, the Cs were calculated as 266, 212, 199, 178, 163, and 158 mAh−1. However, in the abstract, 266 F g−1was denoted. The calculation method should be also provided in the experimental part.

Response to the Reviewer’s comments:

Thank you for your comments. We acknowledge the mistake noted by the reviewer. We have revised the manuscript carefully to omit technical errors. Also, as per the reviewer’s suggestion, we have added the calculation method included in the revised manuscript, and its detailed explanation is included in section 2.7.1.

To put my input together, the revised manuscript is not acceptable. Some polish in writing could be made to make it clearer and crisper.

  1. Figure S3 is missing.

Response to the Reviewer’s comments:

Thank you for your comment. We agree with the reviewer’s comment. As per the reviewer’s suggestion, we have added figure S3 to the revised supporting information.

  1. Table 1 is missing.

Response to the Reviewer’s comments:

Thank you for the reviewer’s comment. According to the reviewer’s suggestions, we have checked and corrected the revised text.

Reviewer 2 Report

The manuscript presents a method for the fabrication of Co2P2O7 supercapacitor electrode material and its evaluation. The paper topic is within the scope of the Materials journal, the conclusions are certainly interesting for the readership of the journal. The paper is structured, the material is presented in a consistent way, and the English is appropriate. For the first time, the authors evaluated the supercapacitors performance of Co2P2O7 nanomaterials prepared hydrothermally at various temperatures. It was found that the treatment temperature governed the structural, morphological, and electrochemical properties of the Co2P2O7. The work provides an advance towards the electrochemical applications of cobalt phosphate nanomaterials, which makes it publishable in the Materials journal upon some major corrections.

1. The novelty of the work has not been sufficiently revealed. In the abstract, text and conclusions of the paper it is necessary to emphasise what has been done by the authors for the first time.

2. The authors claim that a highly porous material with a high surface area was obtained. Unfortunately, the paper does not provide any confirmation of this fact, except for electrochemical measurements. It is possible to confirm high porosity and surface area using low-temperature nitrogen adsorption method? From the SEM data, the material seems to possess very low specific surface area.

3. The X-ray diffraction patterns in Figure 2 are of the very poor quality, making them almost uninformative. These data do not support the conclusion on the single-phase composition of the materials. Please provide XRD patterns of the better quality, with much higher signal-to-noise ratio. Please also analyse the samples in a wider 2 theta range.

4. There are too many terms with “nano-“ prefix in the manuscript title: nanobelts, nanostructure and nanomaterials. One term is enough.

5. The abstract repeatedly states that the obtained cobalt phosphate materials exhibit «higher» properties (specific capacitance and cycling stability). Please indicate, these properties are higher in relation to what exactly?

6. Please, specify the purity of the precursor reagents.

Author Response

Date: 2022-10-19

Response to the Reviewers Comments

Dear Editor,

Materials,

Greetings:

Response letter for manuscript number: R1 (Materials-1958879).

Dear Editor,

This response letter accompanies the online submission of our revised manuscript (Manuscript number: materials-1958879) entitled “A novel synthesized 1D nanobelts-like cobalt phosphate electrode material for excellent supercapacitors application” by S. K. Shinde*, Monali B. Jalak, Swapnil S. Karade, Sutripto Majumder, Mohaseen S. Tamboli, Nguyen Tam Ngu-yen Truong*, Nagesh C. Maile, D. Y. Kim, and Ajay D. Jagadale, for the publication in Materials. We sincerely appreciate the editor and reviewers for reviewing and suggesting valuable comments to improve our manuscript. Following your revision letter, we have thoroughly revised our manuscript to accommodate all the comments and issues raised by the editor and reviewers. We have made substantial modifications to address the reviewer’s suggestions and comments as well as other changes that we thought appropriate to improve our manuscript. All the revisions are accordingly highlighted in this revised manuscript for the publication in  Materials.

Please feel free to contact us for any further information.

Dr. Prof. Nguyen Tam Nguyen Truong

                    M.Sc. Ph. D.               

School of Chemical Engineering, Yeungnam University, 280 Daehak-Ro, Gyeongsan 38541, Republic of Korea

Response letter to reviewers

Reviewer 2; Comments and Suggestions for Authors

The manuscript presents a method for the fabrication of Co2P2O7 supercapacitor electrode material and its evaluation. The paper topic is within the scope of the Materials journal, the conclusions are certainly interesting for the readership of the journal. The paper is structured, the material is presented in a consistent way, and the English is appropriate. For the first time, the authors evaluated the supercapacitors performance of Co2P2O7 nanomaterials prepared hydrothermally at various temperatures. It was found that the treatment temperature governed the structural, morphological, and electrochemical properties of the Co2P2O7. The work provides an advance towards the electrochemical applications of cobalt phosphate nanomaterials, which makes it publishable in the Materials journal upon some major corrections.

The authors are very thankful to the reviewer for appreciating and approving our work. We have addressed all the comments and revised the manuscript thoroughly. The suggested changes have been incorporated into the revised manuscript and the relevant changes are highlighted.

  1. The novelty of the work has not been sufficiently revealed. In the abstract, text and conclusions of the paper it is necessary to emphasise what has been done by the authors for the first time.

Response to the Reviewer’s comments:

Thank you for the reviewer’s comment. As per the reviewer’s suggestion, the abstract section has been thoroughly revised to improve the quality of the manuscript, and its detailed explanation is included in the abstract section on Page no. 2 (line number 1-14).

  1. The authors claim that a highly porous material with a high surface area was obtained. Unfortunately, the paper does not provide any confirmation of this fact, except for electrochemical measurements. It is possible to confirm high porosity and surface area using the low-temperature nitrogen adsorption method? From the SEM data, the material seems to possess a very low specific surface area.

Response to the Reviewer’s comments:

Thank you for your valuable comment. As per the reviewer’s suggestion, we have carried out BET surface area analysis and included BET results in the revised manuscript in the results and discussion section and figure 4. The details about BET surface area analysis are given in the results and discussion section on Pages 5-6 and Figure 4 as follows:

  1. The X-ray diffraction patterns in Figure 2 are of very poor quality, making them almost uninformative. These data do not support the conclusion on the single-phase composition of the materials. Please provide XRD patterns of better quality, with a much higher signal-to-noise ratio. Please also analyze the samples in a wider 2-theta range.

Response to the Reviewer’s comments:

Thank you very much for your valuable comment. We agree to the reviewer’s comment. The XRD pattern was unclear because of the material's very small thickness, nanocrystalline structure, and lower crystallite size. The lower crystallite size and nanocrystalline nature of Co2P2O7 nanobelt-like material show weaker diffraction from the Co2P2O7. Now, we have modified XRD Figure to improve the quality of the peaks and cited some more related research articles. Revised XRD patterns show more distinct peaks and reduce peak noise. Also, from XPS and FT-IR, EDS mapping results we have confirmed the formation of the Co2P2O7. Kindly note that the XRD has been provided to support the formation of Co2P2O7 which is sufficiently confirmed by the XPS and FTIR.

  1. There are too many terms with “nano-“ prefix in the manuscript title: nanobelts, nanostructure and nanomaterials. One term is enough.

Response to the Reviewer’s comments:

We would like to thank the reviewers for carefully reading our manuscript. As Suggested by the reviewer titles have been modified in the revised manuscript.

  1. The abstract repeatedly states that the obtained cobalt phosphate materials exhibit «higher» properties (specific capacitance and cycling stability). Please indicate, these properties are higher in relation to what exactly?

Response to the Reviewer’s comments:

Thank you for reviewer’s comment. As per the reviewer’s suggestion, the abstract section has been thoroughly revised to improve the quality of the manuscript, and its detailed explanation is included in the abstract section on Page no. 1-2.

  1. Please, specify the purity of the precursor reagents.

Response to the Reviewer’s comments:

Thank you for the reviewer’s suggestions. As per the reviewer’s suggestion, the purity of the precursor has been added to the revised manuscript.

Reviewer 3 Report

In this work, the authors have studied hydrothermally Cobalt phosphate nanomaterials at different temperature for their possible application as super capacitors. The results have been well equipped with structural, compositional,  morphological and electorchemical properties of the prepared materials. However, it would be worthy to make the following modifications before acceptance.

1. XRD peaks are not clear at all. It is recommended to do XRD analysis again.

2. If possible to calculate the X-ray density and bulk density and then calculate the percentage porosity for these materials. It will further strengthen your results.

3. Remove spectrum 8, spectrum 6, and spectrum 4 from Figure 3. 

Author Response

Date: 2022-10-19

Response to the Reviewers Comments

Dear Editor,

Materials,

Greetings:

Response letter for manuscript number: R1 (Materials-1958879).

Dear Editor,

This response letter accompanies the online submission of our revised manuscript (Manuscript number: materials-1958879) entitled “A novel synthesized 1D nanobelts-like cobalt phosphate electrode material for excellent supercapacitors application” by S. K. Shinde*, Monali B. Jalak, Swapnil S. Karade, Sutripto Majumder, Mohaseen S. Tamboli, Nguyen Tam Ngu-yen Truong*, Nagesh C. Maile, D. Y. Kim, and Ajay D. Jagadale, for the publication in Materials. We sincerely appreciate the editor and reviewers for reviewing and suggesting valuable comments to improve our manuscript. Following your revision letter, we have thoroughly revised our manuscript to accommodate all the comments and issues raised by the editor and reviewers. We have made substantial modifications to address the reviewer’s suggestions and comments as well as other changes that we thought appropriate to improve our manuscript. All the revisions are accordingly highlighted in this revised manuscript for the publication in  Materials.

Please feel free to contact us for any further information.

Dr. Prof. Nguyen Tam Nguyen Truong

                    M.Sc. Ph. D.               

School of Chemical Engineering, Yeungnam University, 280 Daehak-Ro, Gyeongsan 38541, Republic of Korea

Response to the Reviewer’s # comments

Reviewer #: 3 Comments and Suggestions for Authors

In this work, the authors have studied hydrothermally Cobalt phosphate nanomaterials at different temperatures for their possible application as supercapacitors. The results have been well equipped with structural, compositional, morphological, and electrochemical properties of the prepared materials. However, it would be worth making the following modifications before acceptance.

The authors are very thankful to the reviewer for appreciating and approving our work. We have addressed all the comments and revised the manuscript thoroughly. The suggested changes have been incorporated into the revised manuscript and the relevant changes are highlighted.

  1. XRD peaks are not clear at all. It is recommended to do an XRD analysis again.

Response to the Reviewer’s comments:

Thank you very much for your valuable comment. We agree to the reviewer’s comment. The XRD pattern was unclear because of the material's very small thickness, nanocrystalline structure, and very low crystallite size. The lower crystallite size and nanocrystalline nature of Co2P2O7 nanobelt-like material show weaker diffraction from the Co2P2O7. Now, we have modified XRD Figure to improve the quality of the peaks and cited some more related research articles [1]. Revised XRD patterns show more distinct peaks and reduce peak noise. Also, from XPS and FT-IR, EDS mapping results we have confirmed the formation of the Co2P2O7. Kindly note that the XRD has been provided to support the formation of Co2P2O7 which is sufficiently confirmed by the XPS and FT-IR.

[1] Santosh V Mohite, Ruimin Xing, Bingyue Li, Sanjay S Latthe, Yong Zhao, Xiying Li, Liqun Mao, Shanhu Liu, Spatial compartmentalization of cobalt phosphide in P-doped dual carbon shells for efficient alkaline overall water splitting, Inorg. Chem. 2020, 59, 3, 1996–2004.

  1. If possible calculate the X-ray density and bulk density and then calculate the percentage porosity for these materials. It will further strengthen your results.

Response to the Reviewer’s comments:

Thank you for your valuable comment. We agree with the reviewer’s comment. the XRD pattern shows weak intensity peaks which may be due to the small thickness, nanocrystalline structure, and lower crystallite size of the sample.  If we use the same data for the estimation of X-ray density, bulk density, and porosity percentage, the accuracy of these parameters will be questionable. After changing the deposition temperatures, XRD patterns show the formation of Co2P2O7 crystal structure, but no measurable change in the XRD patterns. Generally depends on the synthesis method and post-treatments or surface treatments. Here, we are more interested in the variation of surface morphologies of Co2P2O7 with the deposition temperature and its consequent effect on the electrochemical supercapacitive properties. Therefore, in order to determine porosity and pore size distribution, a well-known and more accurate BET analysis was performed.

In this paper, we synthesized Co2P2O7 electrode material for electrochemical supercapacitive properties by hydrothermal method. A literature survey shows the various physical and chemical techniques available, out of these, we have chosen hydrothermal. Because the hydrothermal method is presently much more attractive and considerable and does not require sophisticated and other expensive instrumentation. In this paper, we have used different deposition temperatures for developing the nanostructure. To the best of our knowledge, this is the first work that shows how to develop the nanostructure using deposition temperatures. This is the novelty of this paper. And secondly, we have added BET details in the Results and Discussion section.

  1. Remove spectrum 8, spectrum 6, and spectrum 4 from Figure 3.

Response to the Reviewer’s comments:

Thank you for the reviewer’s suggestions. As per the reviewer’s suggestion, we have removed the spectrum world from the revised EDS spectrum.

Round 2

Reviewer 1 Report

Agree to accept.

Author Response

Date: 2022-11-02

Response to the Reviewers Comments

Dear Editor,

Materials,

Greetings:

Response letter for manuscript number: R2 (Materials-1958879).

Dear Editor,

This response letter accompanies the online submission of our revised manuscript (Manuscript number: R2 materials-1958879) entitled “A novel synthesized 1D nanobelts-like cobalt phosphate electrode material for excellent supercapacitors application” by S. K. Shinde, Monali B. Jalak, Swapnil S. Karade, Sutripto Majumder, Mohaseen S. Tamboli, Nguyen Tam Ngu-yen Truong, Nagesh C. Maile, D. Y. Kim, and Ajay D. Jagadale, H. M. Yadav for the publication in Materials. We sincerely appreciate the editor and reviewers for reviewing and suggesting valuable comments to improve our manuscript. Following your revision letter, we have thoroughly revised our manuscript to accommodate all the comments and issues raised by the editor and reviewers. We have made substantial modifications to address the reviewer’s suggestions and comments as well as other changes that we thought appropriate to improve our manuscript. All the revisions are accordingly highlighted in this revised manuscript for the publication in  Materials.

Please feel free to contact us for any further information.

Dr. Prof. Nguyen Tam Nguyen Truong

                    M.Sc. Ph. D.               

School of Chemical Engineering, Yeungnam University, 280 Daehak-Ro, Gyeongsan 38541, Republic of Korea

Response letter to reviewers

Reviewer # 1: Comments and Suggestions for Authors

Agree to accept.

The authors are very thankful to the reviewer for appreciating and approving our work.

Reviewer 2 Report

The authors have changed the text of the manuscript and cleared most of my concerns. 1. However, the quality of the XRD patterns in Figure 2 is basically unchanged. The signal-to-noise ratio does not confirm the presence of only Co2P2O7 phase. 2. Please also comment on the reason for the halo in the diffraction pattern in the low-angle region. I believe that the manuscript can be published in the Materials journal with some minor corrections.

Author Response

Date: 2022-11-01

Response to the Reviewers Comments

Dear Editor,

Materials,

Greetings:

Response letter for manuscript number: R2 (Materials-1958879).

Dear Editor,

This response letter accompanies the online submission of our revised manuscript (Manuscript number: R2 materials-1958879) entitled “A novel synthesized 1D nanobelts-like cobalt phosphate electrode material for excellent supercapacitors application” by S. K. Shinde*, Monali B. Jalak, Swapnil S. Karade, Sutripto Majumder, Mohaseen S. Tamboli, Nguyen Tam Ngu-yen Truong*, Nagesh C. Maile, D. Y. Kim, and Ajay D. Jagadale, H. M. Yadav for the publication in Materials. We sincerely appreciate the editor and reviewers for reviewing and suggesting valuable comments to improve our manuscript. Following your revision letter, we have thoroughly revised our manuscript to accommodate all the comments and issues raised by the editor and reviewers. We have made substantial modifications to address the reviewer’s suggestions and comments as well as other changes that we thought appropriate to improve our manuscript. All the revisions are accordingly highlighted in this revised manuscript for the publication in  Materials.

Please feel free to contact us for any further information.

Dr. Prof. Nguyen Tam Nguyen Truong

                    M.Sc. Ph. D.               

School of Chemical Engineering, Yeungnam University, 280 Daehak-Ro, Gyeongsan 38541, Republic of Korea

Response letter to reviewers

Reviewer 2; Comments and Suggestions for Authors

The authors have changed the text of the manuscript and cleared most of my concerns.

I believe that the manuscript can be published in the Materials journal with some minor corrections.

The authors are very thankful to the reviewer for appreciating and approving our work. We have addressed all the comments and revised the manuscript thoroughly. The suggested changes have been incorporated into the revised manuscript and the relevant changes are highlighted.

Comments 1] However, the quality of the XRD patterns in Figure 2 is basically unchanged. The signal-to-noise ratio does not confirm the presence of only the Co2P2O7 phase.

Response:

Thank you very much for your valuable comment. We agree that the signal-to-noise ratio observed in the XRD pattern is not up to the mark. However, this is mainly due to the material’s crystalline properties. To verify, we executed the XRD run for three attempts as shown in the following figure. The data observed for all these attempts is similar. Noticeably, the characteristic peaks corresponding to the phase Co2P2O7 are clearly distinguished at 24.3, 27.06, 29.72, 35.03, and 43.20° in the samples whereas low intensity denotes the poor crystallinity of all samples. Apart from the phase Co2P2O7 there are no any known obvious phase XRD peaks clearly distinguished in the XRD pattern. Besides, we have provided FTIR, XPS, and SAED analyses which support the formation of the material.

Figure 1 XRD patterns of the CoP-120 and CoP-150 with three different attempts, respectively.

Comments 2] Please also comment on the reason for the halo in the diffraction pattern in the low-angle region.

Response:

Thank you for your comment. The characteristic XRD peaks corresponding to the phase Co2P2O7 come in the range of 20 to 50°. According to the standard JCPDS data, there are some low intense peaks below 20° 2θ angle which are very difficult to distinguish due to noise and poor crystallinity of the samples. Therefore, we have considered only the range of 2θ between 20 to 80°.
